# Function of the Porcine *TRPC1* Gene in Myogenesis and Muscle Growth

**DOI:** 10.3390/cells10010147

**Published:** 2021-01-13

**Authors:** Yu Fu, Peng Shang, Bo Zhang, Xiaolong Tian, Ruixue Nie, Ran Zhang, Hao Zhang

**Affiliations:** 1National Engineering Laboratory for Livestock and Poultry Breeding, Plateau Animal Genetic Resources Center, China Agriculture University, Beijing 100193, China; B20173040294@cau.edu.cn (Y.F.); bozhang0606@cau.edu.cn (B.Z.); 18111010060@fudan.edu.cn (X.T.); s20193040548@cau.edu.cn (R.N.); zhangran0628@cau.edu.cn (R.Z.); 2College of Animal Science, Tibet Agriculture and Animal Husbandry College, Linzhi 860000, China; shangpeng1984@xza.edu.cn

**Keywords:** porcine, *TRPC1*, myogenesis, muscle growth

## Abstract

In animals, muscle growth is a quantitative trait controlled by multiple genes. Previously, we showed that the transient receptor potential channel 1 (*TRPC1*) gene was differentially expressed in muscle tissues between pig breeds with divergent growth traits base on RNA-seq. Here, we characterized *TRPC1* expression profiles in different tissues and pig breeds and showed that *TRPC1* was highly expressed in the muscle. We found two single nucleotide polymorphisms (SNPs) (C-1763T and C-1604T) in *TRPC1* that could affect the promoter region activity and regulate pig growth rate. Functionally, we used RNAi and overexpression to illustrate that *TRPC1* promotes myoblast proliferation, migration, differentiation, fusion, and muscle hypertrophy while inhibiting muscle degradation. These processes may be mediated by the activation of Wnt signaling pathways. Altogether, our results revealed that *TRPC1* might promote muscle growth and development and plays a key role in Wnt-mediated myogenesis.

## 1. Introduction

Muscle development and growth determines the quality of meat and production yield, which are important economic traits in pig farming. Myogenesis, which mainly consists of myofiber formation and hypertrophy, is a complex process that includes myocyte proliferation, migration, differentiation, and fusion to form myotubes, followed by maturation of myofibers [1,2,3]. Although some genes reportedly play important roles in muscle development, the specific regulatory mechanisms remain unclear [4,5,6,7,8].

Transient receptor potential channel 1 (*TRPC1*), the first of the seven mammalian TRP channels to be cloned [9,10], is widely distributed in animal tissues [11]. *TRPC1* is a signaling platform located in the cell and organelle membranes that primarily functions as a nonspecific cation channel within pathways controlling Ca^2+^ entry in response to cell surface receptor activation, thereby affecting cell proliferation and survival [12], differentiation [13], secretion [14,15], and migration [13,16,17]. The *TRPC1*-mediated Ca^2+^ channel is associated with the regulation of physiological functions in several organs and tissues, including the cardiovascular system [18,19,20], central nervous system [21], skeletal and muscular tissues, as well as the immune system [22].

*TRPC1*-siRNA treatment reduced store-operated Ca^2+^ entry and the expression of myogenic differentiation markers, subsequently suppressing skeletal myogenesis [23]. *TRPC1* overexpression could accelerate myoblast fusion and produce hypertrophic myotubes, while *TRPC1* knockdown delayed the onset of Ca^2+^ entry, reducing the speed of cell migration, and affecting the differentiation and fusion of myoblasts [24]. Although mice lacking *TRPC1* could survive until adulthood, they showed defects in skeletal muscle function characterized by a smaller fiber cross-sectional area, less force per cross-sectional area, and less myofibrillar proteins than the controls [25]. In agreement with in vitro experiments, *TRPC1* knockdown affected myoblast differentiation, resulting in smaller myotubes [26]. Furthermore, *TRPC1* may be related to muscle regeneration and Duchenne muscular dystrophy [27,28]. Inhibiting *TRPC1* expression during reloading resists the recovery of muscle mass and slow myosin heavy chain profiles [29]. Calcineurin is involved in the regulation of *TRPC1* expression during muscle regrowth [30]. In summary, *TRPC1* appears to play an important role in muscle development.

Although full-length or partial mRNA sequences have been reported for the seven *TRPC* genes (*TRPC1*-*TRPC7*) in various mammalian species, including humans, mice, rats, rabbits, and bovines [31], most studies on *TRPC1* have mainly focused on humans and mice. The pig *TRPC1* gene is located on chromosome 13 and consists of 12 exons. In pigs, the *TRPC1* gene has been linked with metabolic syndrome and atherosclerosis [32,33]. However, the properties of *TRPC1* and its physiological functions in porcine muscle development are still unknown. Our previous study showed that *TRPC1* was a key gene differentially expressed between pigs with divergent growth phenotypes based on transcriptome and proteome profiles [34]. Therefore, we hypothesized that the pig *TRPC1*, as a candidate gene for growth traits, may be differentially expressed in vivo and may be involved in muscle growth regulation.

Growth rate and body weight are highly variable among pig breeds. Tibetan pig (TP) is a mini-type native Chinese breed that has a slow growth rate and low adult weight [35]. Wujin pig (WJ) is another native Chinese breed that has a slow growth rate and normal adult weight, while the Yorkshire (YY) is a popular commercial breed that has a fast growth rate and normal adult weight [36]. This study investigated the differences in *TRPC1* expression and SNPs among several breeds of pigs (YY; TP; WJ; New Huai population, NHP) and analyzed the association of *TRPC1* with growth traits in these pigs. In addition, we assessed the functions of *TRPC1* on porcine growth traits and explored the possible regulatory pathways. Our findings may provide a theoretical basis for investigating muscle development mechanisms in pigs and other agricultural animals and suggests a novel strategy for agricultural breeding.

## 2. Materials and Methods

### 2.1. Experimental Materials

Ear tissue samples from pigs (WJ, *n* = 29; TP, *n* = 34; YY, *n* = 36; NPH, *n* = 102) were collected for DNA extraction. The TP and WJ are slow-growth indigenous Chinese breeds with small and medium sizes, respectively. YY is the main commercial breed that has characteristics of fast growth and large size. WJ, TP, and YY pigs were raised in Tibet Agriculture and Animal Husbandry College, and the newly bred NHP population was raised in the Anhui Kexin Pig Breeding Farm in Hefei, China. For extraction of total RNA, nine embryos from each group were collected from two pregnant sows 60 days after insemination. The *longissimus dorsi* (LD) muscle tissues from the 12th rib were sampled and snap-frozen in liquid nitrogen and then stored at −80 °C. Animal care was conducted in strict accordance with the Guide for the Care and Use of Laboratory Animals in China. All experiments were approved by the Committee on the Ethics of Animal Experiments of China Agricultural University (permit number: SKLAB-2012-04-07).

### 2.2. DNA, RNA, and cDNA Preparation

Genomic DNA was isolated from ear tissue using the standard phenol/chloroform extraction method, dissolved in a TE solution, and stored at −20 °C. Total RNA was extracted from tissues and cells using TRIzol (Invitrogen, Carlsbad, CA, USA) according to the manufacturer’s instructions and its quality and integrity were verified using a NanoDrop 2000 Biophotometer (Thermo Fisher Scientific Inc, West Palm Beach, FL,USA) and via electrophoresis. RNA samples (2 μg) in a 20 μL reaction volume were reverse transcribed to cDNA using the FastQuant Reverse Transcriptase Kit (TIANGEN, Beijing, China).

### 2.3. Measurement of Gene Expression

Semi-quantitative real-time polymerase chain reaction (SqRT-PCR) was performed as previously described [37], and qRT-PCR was performed using the SYBR Green Master Mix (TIANGEN, Beijing, China) according to the manufacturer’s instructions. Experiments were performed in duplicate, and gene expression levels were calculated using the 2^−ΔΔCt^ method as previously described [38]. We selected a stable housekeeping gene, *β-actin,* as the internal reference. All primers used for this part are presented in Appendix A.

### 2.4. SNP Screening, Genotyping, and Correlation Analysis

Four pairs of primers for SNP screening of the *TRPC1* gene (NC_010455.5) were designed using Primer Premier 5.0 software (Premier Biosoft, Palo Alto, CA, USA) and are listed in Appendix A. The amplicon sequences covered 2201 bp regions in the 5′-flanking and exon 1 of the gene. The PCR products amplified from ten individuals of each group were pooled and sequenced using Sanger sequencing to identify SNPs using Chromas Pro (Technelysium, South Brisbane, Australia) and DNAMAN6.0 (Lynnon, San Ramon, CA, USA). Genotypes of the identified SNPs were determined by individual PCR and Sanger sequencing in the WJ, TP, YY, and NHP populations. The 102 individuals in the NHP population were used to measure age at 30 kg and 90 kg body weight.

### 2.5. Dual-Luciferase Reporter Assays

The promoter regions of *TRPC1* were cloned into a PGL3 luciferase reporter vector (Addgene, Cambridge, Mass, USA). The vector was transfected into a 24-well plate. After 48 h, the cells were lysed in 100 μL of lysis buffer and then assayed for promoter activity using a dual-luciferase reporter assay system (Promega, Madison, WI, USA). The Renilla luciferase signal was normalized to the firefly luciferase signal. The enzymatic activity of luciferase was measured using a PerkinElmer 2030 Multilabel Reader (PerkinElmer, Waltham, MA, USA).

### 2.6. Cell Culture

The C2C12 myoblast cell line (Type Culture Collection of the Chinese Academy of Sciences, Shanghai, China) was cultured at subconfluent densities in a growth medium, made up of Dulbecco’s modified Eagle’s medium (DMEM, Gibco, Grand Island, NY, USA) supplemented with 10% heat-inactivated fetal bovine serum (FBS, Gibco) and 1% penicillin/streptomycin (PS, Gibco). C2C12 cells were differentiated into myocytes or myotubes in a differentiation medium, consisting of DMEM containing 2% horse serum (Gibco) and 1% PS. All cells were maintained in a humidified atmosphere containing 5% CO_2_ at 37 °C.

### 2.7. Plasmids Construction, siRNA Synthesis, and Transfection

For the *TRPC1*-overexpression plasmids, the full-length sequence was cloned into the *Bam*HI (cat. no. R136S, NEB, USA) and *Not*I (cat. no. R0189S, NEB) restriction sites of the expression vector pcDNA3.1. The full-length *TRPC1* sequence was amplified with F/R primers (F: 5′-ATGATGGCGGCCCTGTACCC-3′; R: 5′-TTAATTTCTTGGATAAAACA-3′). *TRPC1* siRNA and scrambled siRNA were synthesized following the manufacturer’s protocol (GenePharma, Shanghai, China). The sequences of *TRPC1* siRNAs are shown in Appendix A. For cell transfection, C2C12 cells and primary myoblasts were transfected with 4 μg of the expression vectors or approximately 10 μL of siRNA oligo using Lipofectamine 2000 (Invitrogen, Carlsbad, CA, USA) in each well of a 6-well plate.

### 2.8. CCK8 and EdU Proliferation Assays

Cell proliferation was measured by CCK8 and 5-ethynyl-2-deoxyuridine (EdU) assays. Cells were cultured in DMEM supplemented with 10% FBS for 24 h. CCK8 (10 µL, Beyotime Biotechnology, Shanghai, China) cells were incubated at 37 °C for 1 h in the dark. The absorbance value was measured using a microplate reader (Biotek, Winooski, VT, USA) at 450 nm. The EdU assay was performed using the EdU assay kit (Ribobio, Guangzhou, China) according to the manufacturer’s instructions. Briefly, 24 or 36 h after transfection, cells were exposed to 50 mM EdU for 2 h. Next, the cells were fixed with 4% paraformaldehyde and permeabilized with 0.5% Triton X-100. Subsequently, cells were incubated in Apollo reaction solution for 1 h and stained with Hoechst 33342 for 30 min. The cells were further analyzed by calculating the ratio of EdU cells to the total number of cells.

### 2.9. Cell Wound Healing and Transwell Migration Assay

Cell migration was measured by the cell wound healing assay described in a previous report [39]. For the transfected cells, a scratch wound was made across the well using a pipette tip. The wounded monolayers were then washed twice with PBS (Gibco) to remove cell debris and incubated with DMEM media with 2% FBS. After 24 h, the wound area and cell migration were photographed under a microscope (ZEISS, Jena, Germany). The transwell migration assay was performed using 24-well transwell chambers (6.5 mm diameter, 8.0 μm pore size; Corning Inc., Corning, NY, USA), in which 10% FBS stimulated cell migration as a chemotactic factor; thus, serum-free cells in the top chamber are more likely to migrate into the bottom chamber with 10% FBS. Briefly, cells were serum-starved with serum-free medium for 6–8 h. Then, cells in 200 µL serum-free medium were placed into the top chamber, and 500 µL complete DMEM was placed into the bottom chamber. After incubation for 12 h, non-migrated cells were removed from the top surface of the transwell with a cotton swab, and the cells that migrated to the lower surfaces of the inserts were stained with 0.1% crystal violet for 0.5–1 h. The migrated cells were photographed under a microscope (ZEISS, Germany). Notably, to avoid the confounding effects of proliferation on cell migration results, the incubation times were markedly less than the doubling times for these cells.

### 2.10. Immunofluorescent Staining

For immunofluorescence, 4% paraformaldehyde-fixed cells were washed with PBS and permeabilized with 0.5% Triton X-100 for 15 min. Cells and sections were supplemented with a blocking buffer (Beyotime Biotechnology, Shanghai, China) at room temperature for 2 h and then incubated with mouse anti-myosin heavy chain (cat. no. M4276, Sigma-Aldrich, St. Louis, MO, USA, 1:500) at 4 °C overnight. Cells were washed briefly and incubated with the fluorescently labeled secondary antibodies (Alexa Fluor 594) for 1 h at room temperature. After washing with PBS, DAPI (Roche Applied Science, Nutley, NJ, USA) was used to stain the nuclei for 5 min. The immunofluorescence images were visualized with a fluorescence microscope (Leica image analysis system, Model Q500MC). The number of nuclei present in one myosin positive cell indicated myoblast fusion.

### 2.11. RNA-seq and Functional Annotation

Total RNA was extracted from control and *TRPC1*-overexpressed C2C12 cells using TRIzol, as mentioned above. cDNA library construction and high-throughput sequencing were performed on the Hiseq-Xten platform by Annoroad Gene Technology Co., Ltd. (Beijing, China). Each group had three biological replicates. The differentially expressed genes (DEGs) were screened using the criterion |log_2_ (fold change)| > 1 and *P*-adjusted ≤ 0.01, and their function was classified using the annotation of Gene Ontology (GO) and Kyoto Encyclopedia of Genes and Genomes (KEGG) pathways with the DAVID online software (http://david.abcc.ncifcrf.gov/home.jsp).

### 2.12. Statistical Analyses

Differences in gene expression, absorbance, and the correlation between the genotypes and traits were analyzed using the one-way analysis of variance (ANOVA) and Student’s *t*-test. Results are expressed as mean ± standard deviation (SD). *P* < 0.05 was considered significant. A χ2 test was used to analyze the distribution of genotypes and to compare differences in genotype distribution.

## 3. Results

### 3.1. TRPC1 is Involved in Growth of Pigs

Expression of the TRPC1 was detected in the heart, LD, liver, back fat (BF), intestine, and leg tissues of the TP pigs (Figure 1A–D). Among these tissues, both the mRNA and protein levels of TRPC1 were higher in the LD, liver, and BF than in other organs, indicating that *TRPC1* is predominantly expressed in tissues involved in growth. The YY pigs had higher *TRPC1* expression in the LD than the TP and WJ pigs (Figure 1E,F), consistent with our previous transcriptomic results [34]. Therefore, we inferred that higher *TRPC1* expression might promote muscle development and growth.

### 3.2. The Genes Flanking TRPC1 Have Two SNPs, C-1763T, and C-1604T

In the 5′-flanking regions of the *TRPC1* gene, two SNPs, C-1763T, and C-1604T, were found (Figure 2A). The two SNPs were linked completely and formed two haplotypes, CC and TT. Three genotypes (CC/CC, CT/CT, and TT/TT) were observed in the WJ and TP and one genotype (CC/CC) in the YY populations, and the distributions of the genotypes conformed to the Hardy–Weinberg equilibrium (*P* > 0.05) (Table 1). YY had higher frequencies of the genotype CC/CC and haplotype CC than the TP and WJ populations (*P* < 0.05). The promoter with the haplotype CC showed higher transcription activity than TT, as evidenced by the dual-luciferase reporter assays (Figure 2B).

The NHP population had three genotypes at the two SNP sites (Table 2). Because only one individual was TT/TT, we compared the age at 30 kg and 90 kg body weights between the CC/CC and CT/CT genotypes. The results showed that the CC/CC genotype exhibited significantly faster growth than the CT/CT genotype (*P* < 0.01) (Table 2).

### 3.3. TRPC1 Promotes Myoblast Proliferation and Migration

One overexpression and four siRNA fragments were constructed, and their detection efficiency was determined. The results showed that siRNA-886 had the highest interference efficiency among the four siRNAs (Figure 3A,B). Thus, *TRPC1* knockdown was treated by siRNA-886 transfection. For proliferation, a lower absorbance value was observed in cells following siRNA-886 transfection relative to the control (Figure 3C), whereas *TRPC1* overexpression markedly improved the absorbance (Figure 3D). *TRPC1* knockdown significantly decreased EdU incorporation (Figure 3E), and *TRPC1* overexpression resulted in increased EdU positivity compared with that of the control (Figure 3F). Consistent with the staining results, *TRPC1* knockdown significantly downregulated the expression of apoptosis markers (*p27, BAD*), and *TRPC1* overexpression showed the same expression pattern (Figure 3G,H). Indeed, the microscopic views showed that the doubling time of cells was ~24 h, and a dramatic reduction in the number of proliferated cells after *TRPC1* knockdown was observed. The opposite effects were observed after *TRPC1* overexpression (Appendix A), highlighting the possibility that *TRPC1* accelerates myoblast proliferation. Besides, the migration capacity was concomitantly attenuated and promoted when *TRPC1* was respectively knocked down and overexpressed (Figure 3I,J).

### 3.4. TRPC1 Promotes Cell Fusion, Differentiation, and Muscle Hypertrophy but Inhibits Muscle Degradation

*TRPC1* expression increased with myoblast differentiation and reached a peak at D4 (Figure 4A); thus, the subsequent differentiation was treated for 4 days (Figure 4B). *TRPC1* overexpression remarkably facilitated myogenic differentiation, as demonstrated by the increased protein expression of myogenic marker MyoG (Figure 4C) and the mRNA level of *MyHC*, *MyoG*, and *MyoD* (Figure 4D), and induced myotube formation (Figure 4F). Meanwhile, the *TRPC1* silenced cells exhibited decreased myogenic marker expression (Figure 4C,E) and formed fewer myotubes (Figure 4G). These results suggest that *TRPC1* positively regulates myoblast differentiation and myotube formation.

Upon overexpression of *TRPC1* mRNA, myoblasts fused to form long, multinucleated MyHC-positive myotubes accompanied by the upregulation of the fusion marker genes Myomaker and *β-1integrin* (Figure 4H,J). In contrast, most C2C12 silencing *TRPC1* remained as short, rounded, and mononucleated myoblasts and downregulated the mRNA levels of fusion markers (Figure 4I,K), suggesting that *TRPC1* facilitates myoblast fusion into one myotube.

Moreover, *TRPC1* overexpression significantly promoted the expression of muscle hypertrophy genes (*Fst* and *Nog*) while inhibiting the expression of muscle degradation markers, including *Atrogin1*, *Bmp4*, *Murf*, and *Foxo3* (Figure 4L). In *TRPC1* knockdown cells, the expression of muscle degradation genes was increased while hypertrophy-related genes were downregulated (Figure 4M). These results indicated that *TRPC1* promoted muscle hypertrophy and suppressed muscle degradation.

### 3.5. Regulative Pathway of TRPC1 on Myogenesis

To identify the regulatory pathway of *TRPC1* in myogenesis, the *TRPC1*-overexpressed and control C2C12 cells after 2 days of culture were subjected to RNA-sequencing. Including *TRPC1*, 1392 DEGs were screened (Appendix A), and enriched GO terms were mainly associated with myotubes or myoblast growth, fusion, and differentiation (Appendix A). The representative KEGG pathways enriched by DEGs mainly contained the Wnt, PI3K-Akt, calcium, and MAPK signaling pathways (Appendix A). Considering key roles in cell proliferation and differentiation, Wnt signaling might be the pivotal pathway underlying the role of *TRPC1* in myogenesis. Several DEGs involved in the Wnt signaling pathway were selected for validation by quantitative real-time polymerase chain reaction (qRT-PCR) (Figure 5A,B). *TRPC1* overexpression upregulated the expression of the Wnt receptor *LRP6* mRNA, downstream *TCF/LEF*, and target genes *CCND*, and downregulated the expression of negative regulators of Wnt (*Axin2*, *GSK3β*, and *SPFR2/4*) (Figure 5A). *TRPC1* knockdown significantly reduced the expression of *LRP6*, *TCF/LEF*, and *CCND* and increased SPFR2/4 expression (Figure 5B). These results suggested that the promoted-growth phenotype of C2C12 cells induced by *TRPC1* might be through the Wnt signaling pathway.

## 4. Discussion

The embryonic development stage is important for postnatal muscle growth. Most commonly, myogenesis and myofibers are formed during the embryonic stage. Approximately 60 days post-insemination, the myofiber formation of porcine embryos is at a critical stage [40,41]. The two Chinese indigenous breeds (TP and WJ) showed slow body growth and small size, whereas an introduced YY breed showed faster growth. This might indicate that YY formed more myofibers than TP and WJ during embryonic development. The higher expression level of *TRPC1* might lead to greater myofiber formation in YY than in TP and WJ, which is consistent with the results of our previous study [34]. Although *TRPC1* exists in multiple tissues, it is mainly expressed in the LD tissue, which indicates that *TRPC1* plays important regulatory roles in skeletal muscle development and facilitates postnatal muscle growth in pigs. High expression in BF implied its fat deposition regulation, which is consistent with previous research [42].

Previous studies have reported that the *TRPC1* gene polymorphisms are associated with type 2 diabetes and diabetic nephropathy in humans [43,44,45,46]. In the present study, we found two SNPs (C-1763T and C-1604T) that formed two haplotypes (CC and TT) in the 5′-flanking region of the *TRPC1* gene in pigs. The genotypes CC/CC and haplotype CC were associated with fast body growth among the breeds (TP, WJ, and YY) and within a population (NHP). The haplotype CC in the promoter region caused a new transcription factor binding site for *Myod1* compared with the TT, as evidenced by an analysis performed with the online software CONSITE (http://consite.genereg.net). The dual-luciferase reporter assays also demonstrated that the haplotype CC would increase transcription activity. Therefore, our results suggest that these two SNPs might regulate the expression of *TRPC1* and postnatal body growth in pigs, and they are potential molecular markers that could be used in pig breeding. Certainly, the regulations of the SNPs on gene expression and muscle development need further to be tested in porcine primary myoblasts.

During myogenesis and muscle regeneration, myoblasts proliferate, migrate, differentiate, and fuse to form multinucleated myotubes that eventually develop into mature muscle fibers [47]. Some studies have demonstrated an essential role of *TRPC1* in myogenesis by interference or knockout methods [13,48,49,50,51]. Our gain and loss of function experiments further confirmed that *TRPC1* induced more proliferated and migrated cells and increased the expression of cell survival markers. *TRPC1^−/−^* mice developed smaller fibers accompanied by decreased expression of myogenic factors [52]. Our results showed that *TRPC1* was more highly expressed in differentiated C2C12 cells, indicating a potential role for myogenic differentiation. Supporting this result, we found that *TRPC1* overexpression enhanced myoblast differentiation and fusion characterized by more polynuclear myotubes and upregulated expression of myogenic and fusion genes. In addition, *TRP* channels have been implicated in muscle hypertrophy and regeneration [53]. We also observed that *TRPC1* promoted muscle hypertrophy but inhibited muscle degradation. Our data collectively suggest that porcine *TRPC1* is a potent regulator of myogenesis and provides a cytological basis for explaining the *TRPC1*-mediated positive regulation on growth rate.

The Wnt/β-catenin signaling plays a crucial role in muscle development in African Cladopus [54], avian [55], and mammals [56], and also affects adult muscle regeneration as well as the proliferation and differentiation of myogenic cell lines in vitro [57,58,59]. Activated Wnt/β-catenin signaling increases muscle mass [60,61] and promotes myoblast proliferation and differentiation [62], while reducing Wnt/β-catenin inhibited myogenic differentiation and the expression of Myomaker, and Myomixer [63]. We observed that *TRPC1* increased transcription of receptors, key factors, and target genes of the canonical Wnt pathway, suggesting that it positively regulates via activation of Wnt signals. The canonical Wnt pathway is only a part of the complex regulatory network for muscle formation. Whether *TRPC1* interacts with noncanonical Wnt signaling or is associated with other signaling pathways obtained from KEGG analysis warrants further study.

In conclusion, our study revealed that *TRPC1* is highly expressed in the muscle of fast-growing pigs and can positively regulate myoblast proliferation, migration, differentiation, fusion, and hypertrophy by activating the Wnt signaling pathway, ensuring myoblasts differentiate into mature muscle fibers, thereby accelerating the growth rate of pigs. We explored the biological function and preliminary regulatory mechanism of *TRPC1* and obtained molecular markers that could be used for genetic improvement. These results provide insights into the molecular mechanism of muscle development, which can be implemented in future breeding programs for pigs.

## Figures and Tables

**Figure 1 cells-10-00147-f001:**
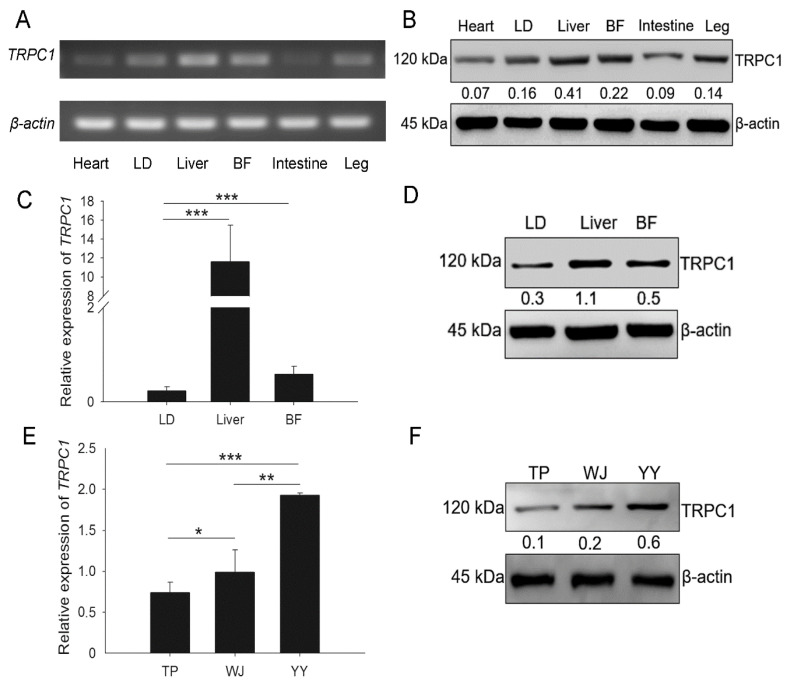
*TRPC1* is involved in the growth of pigs. TRPC1 expression in different tissues of TP pigs at the embryonic stage by SqRT-PCR (**A**), WB (**B**,**D**), and qRT-PCR (**C**). The *TRPC1* mRNA (**E**) and protein (**F**) expression levels in the LD of three pig breeds. LD, *longissimus dorsi*; BF, back fat; TP, Tibetan pig (*n* = 6), WJ, Wujin pig (*n* = 6), YY, Yorkshire (*n* = 6). Each bar represents the means ± SD. * *P* < 0.05, ** *P* < 0.01, *** *P* < 0.001.

**Figure 2 cells-10-00147-f002:**
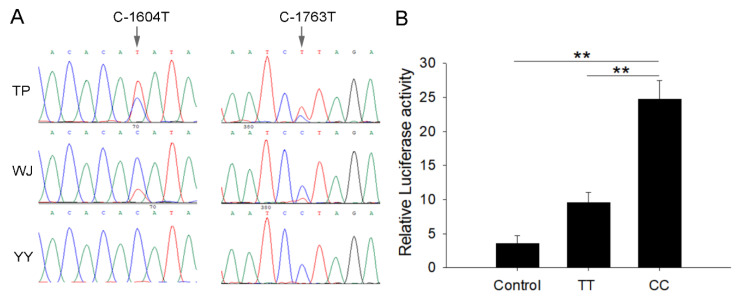
SNP sites and promoter activity analysis. (**A**) The sequencing chromatogram at the sites of two SNPs, C-1604T and C-1763T. The arrows indicate the SNP site. (**B**) Dual-luciferase analysis for promoter activity. Control refers to C2C12 cells co-transfected with PGL3-basic and PRL-TK. CC and TT represent CC and TT homozygous individuals whose promoters are connected to PGL3-basic and co-transfected with PRL-TK cells, respectively. Each bar represents the mean ± SD. ** *P* < 0.01.

**Figure 3 cells-10-00147-f003:**
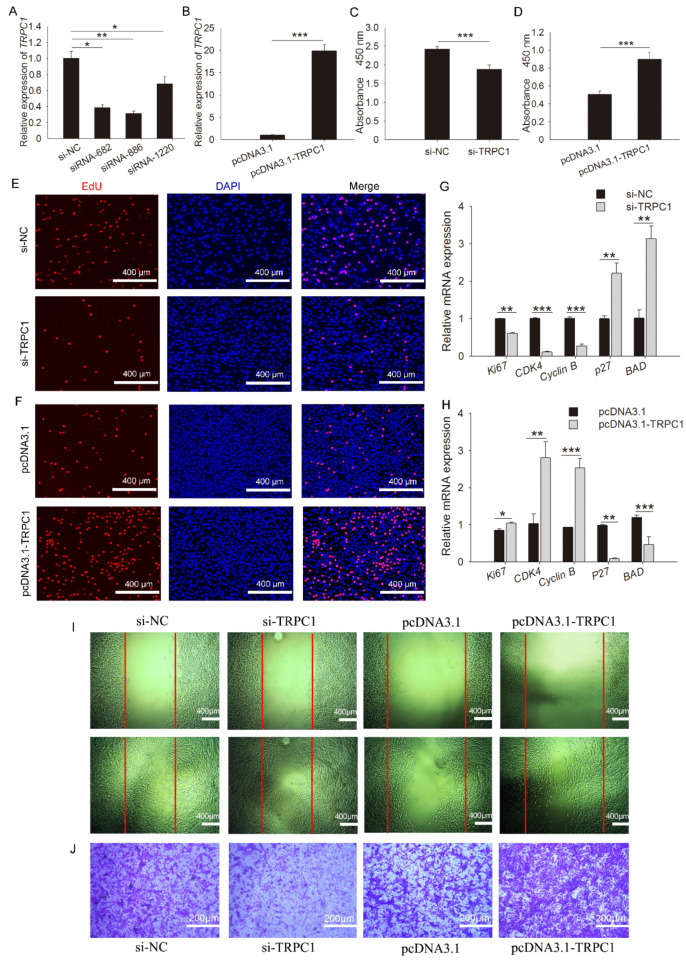
*TRPC1* promoted myoblast proliferation and migration. (**A**) Interference efficiency detection of synthetic siRNA fragments. (**B**) Efficiency detection of plasmid overexpression. (**C**,**D**) CCK8 assay of proliferated myoblasts transfected with RNAi or overexpression fragments. (**E**,**F**) EdU staining for proliferated cells following si-*TRPC1* or pcDNA3.1-*TRPC1* transfection. Nuclei were stained with DAPI, *n* = 3 in each group, scale bar = 400 μm. (**G**,**H**) The mRNA expression levels of proliferation-related genes. (**I**) Wound healing migration assay of C2C12 myoblasts. Some cells were scraped off with a pipette tip to obtain an acellular area. Twenty-four hours later, cells migrated into the acellular area were stained and counted. Cells were migrated in control conditions, Scale bar = 400 μm. (**J**) Transwell test for myoblasts. Purple represents migrated cells, *n* = 3 in each group, scale bar = 200 μm. NC: negative control. The data represent the means ± SD of three independent experiments. * *P* < 0.05, ** *P* < 0.01, *** *P* < 0.001.

**Figure 4 cells-10-00147-f004:**
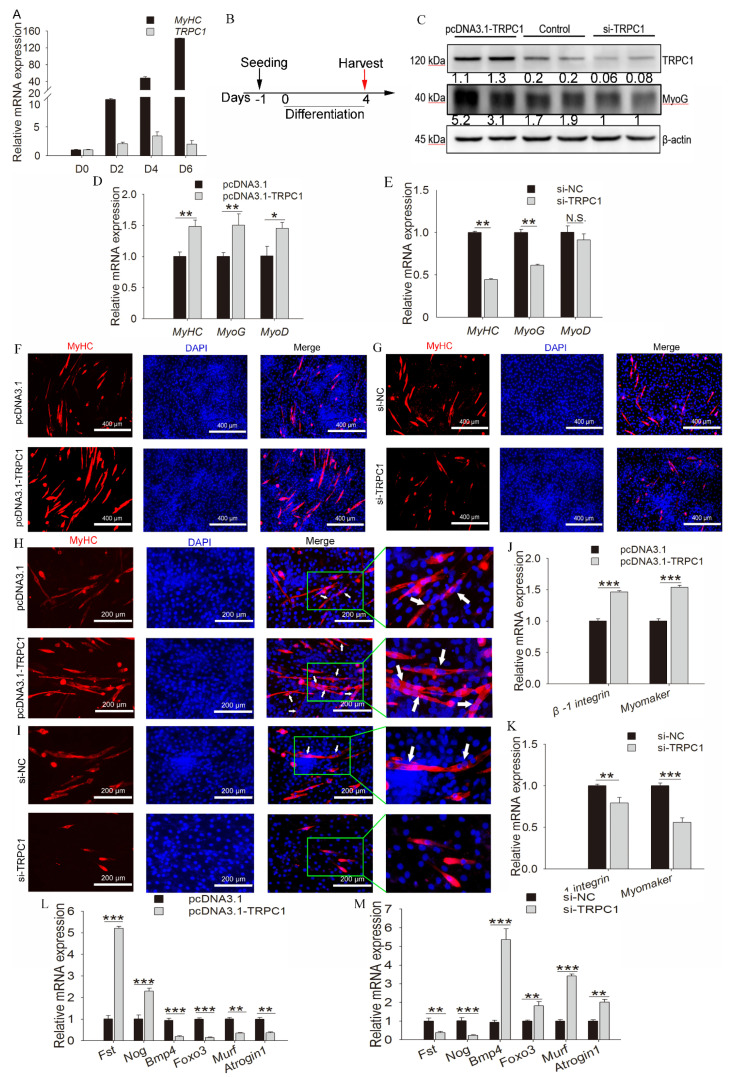
*TRPC1* promoted cell fusion, differentiation, and muscle hypertrophy but inhibited muscle degradation. (**A**) qRT-PCR results showed the expression profiles of the *TRPC1* gene during differentiation. *MyHC* is the myogenic differentiation marker gene. (**B**) Schematic of cell differentiation time. (**C**) The protein level of *TRPC1* and differentiation marker MyoG. (**D**,**E**) The mRNA expression of differentiation marker genes *MyoD*, *MyoG*, and *MyHC* quantified by qRT-PCR. (**F,G**) Immunofluorescence staining for MyHC protein in control or pcDNA3.1-*TRPC1*/si-*TRPC1*-treated myoblasts cultured for 4 days in differentiation medium. MyHC protein expression is shown in red, and nuclei in blue (DAPI). White scale bar = 400 μm. (**H**,**I**) Myoblast fusion analysis by immunofluorescence staining for MyHC, and the arrows represent the multinucleated myotubes, white scale bar = 200 μm. (**J**,**K**) The mRNA expression of fusion marker genes quantified by qRT-PCR. (**L**,**M**) The mRNA expression of muscle degradation markers (*Atrogin1*, *Bmp4*, *Murf*, and *Foxo3*) and muscle hypertrophy genes (*Fst* and *Nog*) quantified by qRT-PCR (*n* = 3, respectively). NC: negative control. The data represent the means ± SD of three independent experiments. N.S.: not significant, * *P* < 0.05, ** *P* < 0.01, *** *P* < 0.001.

**Figure 5 cells-10-00147-f005:**
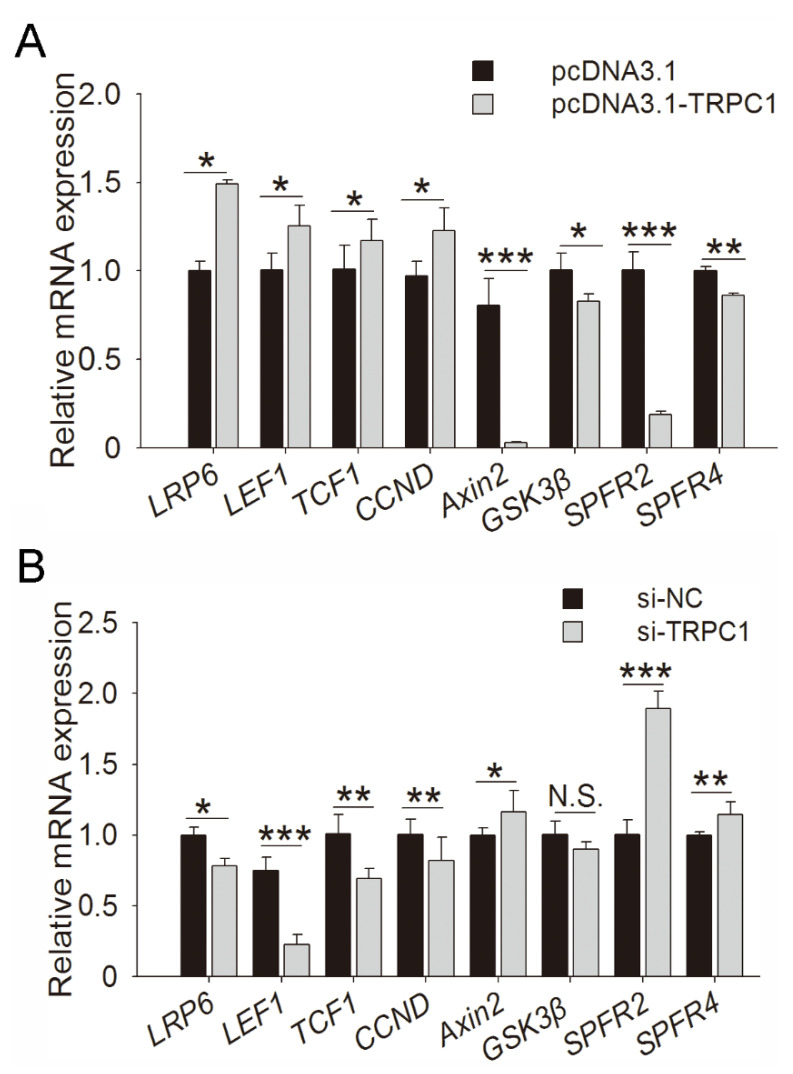
Regulation of *TRPC1* in myogenesis. The mRNA expression of Wnt-related genes in C2C12 cells in which *TRPC1* was overexpressed (**A**) or knocked down (**B**), respectively. NC: negative control. The data represent the means ± SD of three independent experiments. N.S.: not significant, * *P* < 0.05, ** *P* < 0.01, *** *P* < 0.001.

**Table 1 cells-10-00147-t001:** Genotype and haplotype frequencies in the C-1763T and C-1604T in pig populations.

Breed	Sample Size	Haplotype Frequency		Genotype Frequency			χ2 Value (*P*-Value)
		CC	TT	CC/CC	CT/CT	TT/TT	
TP	34	0.441	0.559	0.235	0.412	0.353	1.275 (0.529)
WJ	29	0.621	0.379	0.379	0.483	0.138	0.202 (0.904)
YY	36	1	0	1	0	0	/

**Table 2 cells-10-00147-t002:** Analysis of the effects of CC, CT, and TT genotypes of the *TRPC1* gene in New Huai pigs.

Days to:	CC/CC (*n* = 80)	CT /CT (*n* = 21)	TT/TT (*n* = 1)
30 kg	92.45 ± 0.97 *	101.04 ± 2.49	128.66
90 kg	189.27 ± 4.30 *	209.31 ± 6.85	209.93

* Statistically significant difference between CC/CC and CT/CT groups (*P* < 0.01). Statistical significances of TT/TT groups with only one individual were not analyzed.

## Data Availability

The original data in this study are openly available in Dataverse Project, reference number [GSE160972].

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
