# Peer review of "Function of the Porcine TRPC1 Gene in Myogenesis and Muscle Growth"

_cells, 2021, doi:10.3390/cells10010147_

Round 1
Reviewer 1 Report
In their manuscript Fu et al put in evidence the expression profile of TRPC1 and its SNPs in several swine breeds. The work presented is original and pertinent not only for the TRP channel field but also for its agricultural perspectives. Indeed, the authors correlate TRPC1 expression with the muscle growth and development mechanisms in pigs and suggest the implication of Wnt-mediated myogenesis.
Overall, I think this a well-designed and sound study. I propose here below some points that
would improve the manuscript.
- It would be helpful for the reader to have the information on body growth and size of the pig breeds used in the study in the introduction and not only in the discussion in order to better follow the manuscript.
- The protein expression tissue profile of the different breeds is needed in addition to the transcript profiling
- It would be interesting to introduce the identified SNPs, C-1763T and C-1604T, and test their effect in cell fusion, differentiation, muscle hypertrophy and muscle degradation in vitro.
Author Response
Response to Reviewer 1 Comments
Point 1: It would be helpful for the reader to have the information on body growth and size of the pig breeds used in the study in the introduction and not only in the discussion in order to better follow the manuscript.
Response: We have added the description in the Introduction section. (Lines 62-65)
Point 2: The protein expression tissue profile of the different breeds is needed in addition to the transcript profiling
Response: Thank you for your suggestion. We have added the protein expression in Fig. 1 and have added the corresponding description. (Lines 185-191)
Point 3: It would be interesting to introduce the identified SNPs, C-1763T and C-1604T, and test their effect in cell fusion, differentiation, muscle hypertrophy and muscle degradation in vitro.
Response: Thank you for your suggestions. We attempted to test the roles of the SNPs on cell differentiation, fusion, muscle hypertrophy, and degradation in vitro. However, due to the influence of African swine fever and COVID-19, we were unable to obtain primary porcine myoblasts. To explain functions of the SNPs, we measured promotor activity, and the results indicated that the CC haplotype upregulated gene expression more than the TT haplotype. Furthermore, the effects of different TRPC1 expressions on cell process were detected in C2C12 cells. Taken together, these results indirectly show that the TRPC1 gene and the identified SNPs play an important regulatory role in muscle development.
Special thanks to you for your good comments.
Reviewer 2 Report
Fu et al. have suggested that TRPC1 might promote muscle growth and development and it plays a role in Wnt-mediated myogenesis. They suppose that TRPC1 is highly expressed in porcine muscle cells and it can positively regulate myoblast proliferation, migration, differentiation, fusion, and hypertrophy by the activation of Wnt signaling pathway.
Main Comment.
The present study can be divided in two parts. In the first, authors demonstrate that TRPC1 is overexpressed in fast growing pigs, as Yorkshire pigs, compared to slower growing pigs. Authors also demonstrate that TRPC1 is predominantly expressed in tissues involved in growth. In addition, authors identified two SNP on the 5’-flanking regions of TRPC1 gen linked to TRPC1 expression and the growth rate of the pigs. Altogether, these results suggest that higher TRPC1 expression might promote muscle development and growth, as they conclude in the manuscript.
In the second part of the manuscript, authors demonstrate the role of TRPC1 in myoblast proliferation, migration, cell fusion, differentiation or muscle hypertrophy, and they suggest the involvement of Wnt/β-catenin signaling pathway in these TRPC1- mediated processes. For that, authors use the C2C12 myoblast cell line (mouse myoblast cells), in which the expression of TRPC1 was genetically modified by the use of specific siRNAs against the TRPC1 mRNA to abolish its expression or by the use of a plasmid to induce TRPC1 overexpression. However, the originality and novelty of this part of the study is very poor since the role of TRPC1 in the above-mentioned processes has been described in different studies (Louis et al, 2008 (PMID: 19001499), Formigli et al, 2009 PMID: 19351713; Antigny et al, 2013 PMID: 23549783; Xia et al 2016 PMID: 26752511). The only new data is the possible involvement of Wnt signaling pathway TRPC1-mediated processes, although it need to be further investigated in order to arise this conclusion.
Other comments.
- Authors demonstrate TRPC1 expression by PCR in porcine and murine cells. It would necessary to detect the protein expression by western blotting.
- Language should be revised, there are mistakes throughout the text that make it difficult to read and understand.
- Bibliography should be checked since there are different citation styles.
Author Response
Response to Reviewer 2 Comments
Point 1: The originality and novelty of this part of the study is very poor since the role of TRPC1 in the above-mentioned processes has been described in different studies (Louis et al, 2008 (PMID: 19001499), Formigli et al, 2009 PMID: 19351713; Antigny et al, 2013 PMID: 23549783; Xia et al 2016 PMID: 26752511).
Response: Thank you for comments. For verifying the roles of porcine TRPC1, we performed a cytological test that has been partly described in previous studies. The major difference of our study from these previous studies is that the overexpressed gene was porcine TRPC1 in our study. In addition, we verified the function of TRPC1 via gain and loss experiments on cell proliferation (EdU and CCK8 assays), cell migration (wound healing and Transwell assays), cell differentiation, cell fusion, and muscle hypertrophy and degradation. We think that our study comprehensively demonstrated the role of TRPC1 in muscle development.
Point 2: Authors demonstrate TRPC1 expression by PCR in porcine and murine cells. It would necessary to detect the protein expression by western blotting.
Response: Thank you for your suggestion. We have added the protein expression in Fig. 1 and Fig. 4 and have included the corresponding description. (Lines 185-191, 250-278)
Point 3: Language should be revised, there are mistakes throughout the text that make it difficult to read and understand.
Response: Thank you for your comments. Our manuscript was edited by an English language editing company.
Point 4: Bibliography should be checked since there are different citation styles.
Response: We have checked and revised the bibliography in the revised manuscript.
Special thanks to you for your good comments.
Reviewer 3 Report
In the present report Fu et al. have investigated putative importance of TRPC1 as a inducer of muscle growth and development pigs. They have investigated several breeds of pigs. In their investigation they have used molecular biology methods, and cultured muscle cells. The manuscript is well written and, in my opinion, easy to follow. As such the investigation is interesting, although it would have been interesting to relate the overexpression and knock out data in the in vitro experiments to e.g. calcium signaling and the possible importance in muscle development in the pig breed. I have some comments for the Authors to consider.
- the Authors incubated the C2Cl2 myoblast cells for 24 h in the proliferation assay, and the wound healing and transwell assays were also conducted for 24 h. The Authors mention that this was not a problem, as the doubling time for the cells is much less. Please provide information on doubling time. In Fig XX, the CCK8 assay shows an almost doubling in the TRPC-overexpressing cells. This suggests that
- In Figure 1 the Authors only show mRNA levels – what about the actual protein levels?
- In the legend of Figure 2B, please be more precise in regard to breed.
- Line 226 in the manuscript. The Authors are talking about “trends” regarding the results in Figure 3. As the results clearly are significant, the word “trends” is not necessary! Furthermore, in Fig 3E it is a bit confusing as the Dapi staining of siTRPC1 cells is much stronger (although the EdU is less strong) than the control. Please explain.
- In Figure 4 at least I have difficulties seeing the multinucleated cells…probably due to the small image…
- Please check the references. Why are they numbered in the reference list, but used as first Author in the text? Furthermore, reference 22 and42 lack the journal.
Author Response
Response to Reviewer 3 Comments
Point 1: The Authors incubated the C2Cl2 myoblast cells for 24 h in the proliferation assay, and the wound healing and transwell assays were also conducted for 24 h. The Authors mention that this was not a problem, as the doubling time for the cells is much less. Please provide information on doubling time. In Fig XX, the CCK8 assay shows an almost doubling in the TRPC-overexpressing cells.
Response: Thank you for your comments. The doubling time of cell proliferation in 10% FBS was ~24 h (Fig. S1), which has been added to the revised manuscript (Line 229). To avoid the effect of proliferation, the wounded monolayers were incubated with DMEM with 2% FBS instead of 10% FBS which was used for proliferation (Lines 151-152). In the transwell assay, 10% FBS acted as a chemotactic factor of cell migration and was used in the bottom chamber. Thus, we detected the cell migration after incubation for 12 h, in which time most cells migrated rather than proliferated. (Lines 154-156).
Point 2: In Figure 1 the Authors only show mRNA levels – what about the actual protein levels?
Response: We have added the protein expression in Fig. 1 and the corresponding description to the revised manuscript. (Lines 185-191)
Point 3: In the legend of Figure 2B, please be more precise in regard to breed.
Response: In Fig. 2B, CC and TT represent the promoter regions of TRPC1 with the CC and TT haplotype in the two SNPs (C-1604T and C-1763T), respectively. The sequences were cloned from pig individuals that have a CC/CC or TT/TT genotype. Except for the two SNP sites, the sequences were the same among the TP, WJ, and YY breeds.
Point 4: Line 226 in the manuscript. The Authors are talking about “trends” regarding the results in Figure 3. As the results clearly are significant, the word “trends” is not necessary! Furthermore, in Fig 3E it is a bit confusing as the Dapi staining of siTRPC1 cells is much stronger (although the EdU is less strong) than the control. Please explain.
Response: Thank you for your suggestions. We have modified the sentences (Line 227-229). The stronger DAPI staining of siTRPC1 cells might have been caused by the uneven number of initial cells. Although the siTRPC1 groups had more initial cells, the red positive EdU cells showed lower proliferative ability in the siTRPC1 groups than those in the control. The results still indicated that the proliferative ability of the interference group was lower than that of the control.
Point 5: In Figure 4 at least I have difficulties seeing the multinucleated cells…probably due to the small image…
Response: We have enlarged the image in the Fig. 4.
Point 6: Please check the references. Why are they numbered in the reference list, but used as first Author in the text? Furthermore, reference 22 and42 lack the journal.
Response: We have checked and revised the references in the revised manuscript.
Special thanks to you for your good comments.
Round 2
Reviewer 1 Report
The authors have responded to my comments eventhough the experiments with the SNPs were not performed due to the influence of African swine fever and COVID-19. I would propose that they pit it as perspectives in their discussion section.
Author Response
Response to Reviewer 1 Comments
Point 1: The authors have responded to my comments eventhough the experiments with the SNPs were not performed due to the influence of African swine fever and COVID-19. I would propose that they pit it as perspectives in their discussion section.
Response: Thank you for your suggestion. We have added the perspective description in the discussion section. (Lines 330-331)
Special thanks to you for your good comments.

Reviewer 2 Report
Thank you for comments. For verifying the roles of porcine TRPC1, we performed a cytological test that has been partly described in previous studies. The major difference of our study from these previous studies is that the overexpressed gene was porcine TRPC1 in our study. In addition, we verified the function of TRPC1 via gain and loss experiments on cell proliferation (EdU and CCK8 assays), cell migration (wound healing and Transwell assays), cell differentiation, cell fusion, and muscle hypertrophy and degradation. We think that our study comprehensively demonstrated the role of TRPC1 in muscle development.
I recognize the efforts of the authors to improve the quality and novelty of the manuscript. It is true that the porcine gene is overexpressed in this study, but in my opinion, it is not novel enough since similar results were observed with the overexpression of the human gene. It would be convenient to see if there are differences in the effect of overexpression of the human gene in your model and experimental method compare to porcine gene. Regarding the silencing experiments of TRPC1 expression in the C2C12 myoblast cell line, I suppose that used siRNAs block the expression of native TRPC1 (murine gene), so the observed effects are due to the loss of expression of native TRPC1 and not purine TRPC1. To achieve this objective, author should overexpress a negative dominant of porcine TRPC1 (as the human TRPC1 (F562A)).
Author Response
Response to Reviewer 2 Comments
Point 1: I recognize the efforts of the authors to improve the quality and novelty of the manuscript. It is true that the porcine gene is overexpressed in this study, but in my opinion, it is not novel enough since similar results were observed with the overexpression of the human gene. It would be convenient to see if there are differences in the effect of overexpression of the human gene in your model and experimental method compare to porcine gene. Regarding the silencing experiments of TRPC1 expression in the C2C12 myoblast cell line, I suppose that used siRNAs block the expression of native TRPC1 (murine gene), so the observed effects are due to the loss of expression of native TRPC1 and not purine TRPC1. To achieve this objective, author should overexpress a negative dominant of porcine TRPC1 (as the human TRPC1 (F562A)).
Response: Thank you for your valuable suggestions. In the siRNAs experiment, the observed effects were really due to the loss of expression of murine TRPC1, and the effects of silencing expression of porcine TRPC1 are just inferred based on functional similarity of homologous genes in animals. No mutation in coding region of the porcine TRPC1 was found between pig breeds with divergent growth phenotypes. We also do not know silent mutations in the porcine TRPC1 gene, so we have no way to overexpress of a negative dominant of porcine TRPC1 (as the human TRPC1 (F562A, D639K, or D640K)). Relative to the previous studies (Min et al, 2009; Kim et al, 2003; Joseph et al, 2003), our study tested various cytology behaviors in C2C12 cells to explain functions of TRPC1 on muscle development in pigs.
References:
Kim S.J., Kim Y.S., Yuan J.P.; et al. Activation of the TRPC1 cation channel by metabotropic glutamate receptor mGluR1. Nature. 2003, 426, 285-291.
Joseph P.Y., Kirill K., Dong M.S.; et al. Homer binds TRPC family channels and is required for gating of TRPC1 by IP3 receptors. Cell. 2003, 114, 777-789.
Kim M.S., Zeng W., Yuan J.P.; et al. Native stor-operated Ca2+ influx requires the channel function of Orai1 and TRPC1. J Biol Chem. 2009, 284, 9733-9741.

Round 3
Reviewer 2 Report
Authors have made an effort to improve and defend the quality of their work. Like my other fellow reviewers, I will accept the publication of the article in the present form.